Urinary complement proteins in IgA nephropathy progression from a relative quantitative proteomic analysis

Niu Xia 1
Zhang Shuyu 2
Shao Chen 1
Guo Zhengguang 1
http://orcid.org/0000-0001-6773-9289 Wu Jianqiang 3
Tao Jianling 2
Zheng Ke 2
Ye Wenling 2
Cai Guangyan 4
Sun Wei 1 sunwei@ibms.pumc.edu.cn
http://orcid.org/0000-0002-1876-4186 Li Mingxi 2 mingxili@hotmail.com
1 Core Facility of Instruments, Institute of Basic Medical Sciences, Chinese Academy of Medical Sciences, School of Basic Medicine, Peking Union Medical College , Beijing , China
2 Department of Nephrology, State Key Laboratory of Complex Severe and Rare Diseases, Peking Union Medical College Hospital, Chinese Academy of Medical Science and Peking Union Medical College , Beijing , China
3 Medical Research Center, Peking Union Medical College Hospital, Chinese Academy of Medical Sciences and Peking Union Medical College , Beijing , China
4 Department of Nephrology, The First Medical Centre, Chinese PLA General Hospital, Medical School of Chinese PLA , Beijing , China
Capusa Cristina
Electronic publication date: 2023 Apr 11
Publication date: 2023
Volume: 11
Electronic Location ID: e15125
Received 2021 Dec 23; Accepted 2023 Mar 3
Copyright: © 2023 Niu et al.
Copyright year: 2023
Copyright holder: Niu et al.
License: This is an open access article distributed under the terms of the Creative Commons Attribution License, which permits unrestricted use, distribution, reproduction and adaptation in any medium and for any purpose provided that it is properly attributed. For attribution, the original author(s), title, publication source (PeerJ) and either DOI or URL of the article must be cited.
License URL: https://creativecommons.org/licenses/by/4.0/

Keywords: Complement proteins, IgA nephropathy, α-N-acetylglucosaminidase, Proteomics, Urine

Funding: National Natural Science Foundation of China 31200614 Beijing Municipal Science and Technology Project D181100000118004 Youth Fund of Peking Union Medical College Hospital pumch201912152 This work was supported by the National Natural Science Foundation of China (No. 31200614), the Beijing Municipal Science and Technology Project (D181100000118004), and the Youth Fund of Peking Union Medical College Hospital (pumch201912152). The funders had no role in study design, data collection and analysis, decision to publish, or preparation of the manuscript.

==============================
Aim

IgA nephropathy (IgAN) is one of the leading causes of end-stage renal disease (ESRD). Urine testing is a non-invasive way to track the biomarkers used for measuring renal injury. This study aimed to analyse urinary complement proteins during IgAN progression using quantitative proteomics.

Methods

In the discovery phase, we analysed 22 IgAN patients who were divided into three groups (IgAN 1-3) according to their estimated glomerular filtration rate (eGFR). Eight patients with primary membranous nephropathy (pMN) were used as controls. Isobaric tags for relative and absolute quantitation (iTRAQ) labelling, coupled with liquid chromatography-tandem mass spectrometry, was used to analyse global urinary protein expression. In the validation phase, western blotting and parallel reaction monitoring (PRM) were used to verify the iTRAQ results in an independent cohort (N = 64).

Results

In the discovery phase, 747 proteins were identified in the urine of IgAN and pMN patients. There were different urine protein profiles in IgAN and pMN patients, and the bioinformatics analysis revealed that the complement and coagulation pathways were most activated. We identified a total of 27 urinary complement proteins related to IgAN. The relative abundance of C3, the membrane attack complex (MAC), the complement regulatory proteins of the alternative pathway (AP), and MBL (mannose-binding lectin) and MASP1 (MBL associated serine protease 2) in the lectin pathway (LP) increased during IgAN progression. This was especially true for MAC, which was found to be involved prominently in disease progression. Alpha-N-acetylglucosaminidase (NAGLU) and α-galactosidase A (GLA) were validated by western blot and the results were consistent with the iTRAQ results. Ten proteins were validated in a PRM analysis, and these results were also consistent with the iTRAQ results. Complement factor B (CFB) and complement component C8 alpha chain (C8A) both increased with the progression of IgAN. The combination of CFB and mucosal addressin cell adhesion molecule-1 (MAdCAM-1) also showed potential as a urinary biomarker for monitoring IgAN development.

Conclusion

There were abundant complement components in the urine of IgAN patients, indicating that the activation of AP and LP is involved in IgAN progression. Urinary complement proteins may be used as biomarkers for evaluating IgAN progression in the future.

Introduction

IgA nephropathy (IgAN) is the most common primary glomerulonephropathy worldwide, and 20–30% of patients with IgAN progress to end-stage renal disease within 10 years of disease onset (Floege & Amann, 2016). The “four-hit hypothesis” of IgAN pathogenesis (Suzuki et al., 2011) involves: elevated serum galactose-deficient IgA1 (Gd-IgA1), IgG or IgA antibodies specific to Gd-IgA1, the formation of immune complexes (ICs), and IC deposits predominantly on the glomerular mesangium, leading to inflammation, complement activation, mesangial hypercellularity, and glomerulosclerosis.

Activation of the complement cascade is known to be involved in the pathogenesis of IgAN. The alternative and lectin pathways dominate the complement activation of IgAN (Tortajada et al., 2019). Data from mass spectrometry (MS) analyses of kidney biopsy samples have demonstrated that the complement pathways are involved in IgAN (Paunas et al., 2017), but previous studies have not revealed the role of complement proteins in IgAN progression.

Because urine is non-invasively accessible, has a low background, and is a relatively stable biofluid, it is an ideal source for biomarker discovery using proteomics. Urinary proteomics could be used to analyse protein profiles and assess pathogenetic mechanisms involved in disease progression. Some previous studies using urinary proteomics in IgAN focused on candidate biomarker discovery (Prikryl et al., 2017; Guo et al., 2018), but there has not yet been a study on urinary proteomic profile changes during IgAN progression.

A proteomic study usually includes two phases: discovery and validation. In the discovery phase, the samples are analysed using an untargeted proteomics approach to identify the differentially expressed proteins, which may be related to disease mechanisms, drug targets, or disease biomarkers (Wu et al., 2021). In the validation phase, the differential proteins identified in the discovery phase are then analysed using targeted methods, including western blotting, ELISA, and parallel reaction monitoring (PRM), usually in another cohort to validate the findings of the discovery phase (Wilhelm et al., 2014).

Isobaric tags for relative and absolute quantification (iTRAQ) is a quantitative proteomics technique used to identify and quantify protein expression levels. PRM can distinguish between interference information and real signals and has greater selectivity in detecting target proteins, so it is often used to verify the differential proteins identified using iTRAQ.

In this study, urine samples were collected from IgAN patients with different disease severities and from primary membranous nephropathy (pMN) patients. Fourteen highly abundant proteins were first removed from the urine samples of IgAN and pMN patients, and then the samples were analysed using iTRAQ labelling coupled with liquid chromatography-tandem mass spectrometry (2D-LC/MS/MS) for the quantitative proteome analysis. PRM was then used to validate the differentially expressed proteins. The aim of this study was to analyse the entire urinary protein profiles of IgAN patients at different disease stages to assess whether the overall urinary complement expression profile changes with disease progression.

Materials and Methods

Reagents and instruments

HPLC-grade acetonitrile (ACN), trifluoroacetic acid, formic acid, ammonium bicarbonate, iodoacetamide (IAA), and dithiothreitol (DTT) were all purchased from Sigma (St. Louis, MO, USA). Sequencing-grade trypsin was purchased from Promega (Madison, WI, USA). The 4-plex iTRAQ reagents were purchased from ABsciex (Framingham, MA, USA). An Orbitrap Fusion Lumos Tribrid mass spectrometer, coupled with an EASY-nLC 1000 HPLC system, was purchased from Thermo Fisher Scientific (Waltham, MA, USA) and used for the proteome analysis in both the discovery and validation phases. Primary antibodies against α-N-acetylglucosaminidase (NAGLU, ab72178) and α-galactosidase A (GLA, ab168341) were purchased from Abcam (Cambridge, UK) for western blotting.

Patients and urine samples

All the patients enrolled in this study were diagnosed with IgAN by kidney biopsy. Fresh urine samples were collected on the morning of kidney biopsy. For the discovery cohort, the sample collection duration was from August 2014 to January 2017, and the sample collection for the validation cohort was from November 2019 to July 2021. The urine was centrifuged at 3,000×g for 30 min. The sediment was discarded, and the supernatant was stored at −80 °C. This study was approved by the Ethics Committee of Peking Union Medical College Hospital (approval number: JS-2573), and the enrolled patients signed informed consent forms. The amount of 24-h urine proteins (24hUP), serum creatinine (SCr), and estimated glomerular filtration rate (eGFR) of the patients were evaluated. The CKD-EPI equation was used to estimate the eGFR (Levey et al., 2009). The clinical characteristics of the patients in the discovery and validation phases are shown in Table 1. The pMN was diagnosed by renal pathology and clinical correlation as the Brenner/Rector’s The Kidney, 9th proposed (Saha et al., 2011). According to the histological diagnoses: no significant mesangial hypercellularity observed by light microscopy, predominant IgG4 subclass along GBM and mesangial Ig staining absent by immunofluorescence, and mesangial electron-dense deposits absent by electron microscopy, as well as the clinical investigations for secondary factor (e.g., infection, lupus, drugs, malignant neoplasms, and other causes), we basically ruled out the occurrence of secondary MN in all subjects. Moreover, the serologic test for anti-PLA2R was absent for the diagnosis because it was not feasible in our hospital when the urine samples were collected. In both the discovery and validation phases, IgAN patients were divided into three subgroups based on eGFR: IgAN-1 (eGFR: >90 ml/min/1.73 m2), IgAN-2 (eGFR: 60–90 ml/min/1.73 m2), and IgAN-3 (eGFR: 30–60 ml/min/1.73 m2). Two nephropathologists evaluated the renal biopsy samples independently according to the Oxford Classification of IgA nephropathy from the IgA Nephropathy Classification Working Group in 2016 (Trimarchi et al., 2017).

Table 1 Clinical and histopathological characteristics of patients in discovery and validation phase.

Group	N	Gender (M/F)	Age
(year)	24 h UP
(g)	SCr
(mmol/L)	eGFR
(ml/min/1.73 m2)	C3
(g/L)	C4
(g/L)	Histopathological characteristic	
Discovery phase	
IgAN-1	8	5/3	33.25 ± 6.86	1.36 ± 0.84	74.10 ± 13.62	112.20 ± 11.55	1.05 ± 0.16	0.21 ± 0.04	M1E0-1S0-1T0-C0-1	
IgAN-2	5	4/1	39.20 ± 12.66	2.09 ± 1.25	103.60 ± 12.46a	74.78 ± 11.32a	0.94 ± 0.08	0.24 ± 0.06	M1E0-1S1T0-2-C1	
IgAN-3	9	7/2	38.11 ± 11.92	1.78 ± 0.73	168.44 ± 59.61a,b	45.37 ± 11.76a,b	1.09 ± 0.33	0.24 ± 0.09	M1E0-1S1T0-2-C0-1	
MN	8	4/4	40.50 ± 11.65	2.86 ± 0.92a,c	68.63 ± 17.28b,c	115.62 ± 21.99b,c	1.18 ± 0.21.	0.26 ± 0.09	MN I-II	
WB validation	
IgAN-1	6	1/5	35.83 ± 10.19	1.06 ± 0.58	63.17 ± 11.99	116.78 ± 11.24	1.04 ± 0.21	0.22 ± 0.03	M1E0-1S0-1T0-C0-1	
IgAN-2	8	6/2	38.38 ± 8.07	1.45 ± 1.00	105.63 ± 15.84a	72.47 ± 7.86a	1.07 ± 0.20	0.24 ± 0.07	M1E1S1T0-2-C1	
IgAN-3	3	3/0	39.00 ± 3.46	1.69 ± 0.26	159.00 ± 17.06a,b	47.01 ± 7.22a,b	1.19 ± 0.58	0.23 ± 0.07	M1E0-1S1T2-C1	
MN	5	2/3	47.60 ± 5.41	1.59 ± 1.14	63.20 ± 10.99b,c	115.5 ± 19.44b,c	1.25 ± 0.19	0.27 ± 0.12	MN I-II	
PRM validation	
IgAN-1	14	8/6	35.57 ± 12.8	1.48 ± 0.69	73.14 ± 10.42	110.73 ± 22.67	1.12 ± 0.18	0.21 ± 0.09	M0-1E0-1S0-1T0-1-C0-1	
IgAN-2	29	17/12	39.24 ± 11.63	1.86 ± 1.11	102.72 ± 18.67a	71.77 ± 8.50a	1.13 ± 0.16	0.23 ± 0.06	M0-1E0-1S0-1T0-2-C0-1	
IgAN-3	21	12/9	45.14 ± 13.35	2.13 ± 1.15	128.29 ± 16.60a	52.00 ± 4.47a,b	1.12 ± 0.14	0.24 ± 0.06	M0-1E0-1S0-1T0-2-C0-2	
Note:

N, number; 24 h UP, 24 h urinary protein; SCr, serum creatinine; C3, complement 3; C4, complement 4. The five pathological variables in Oxford classification were scored as follow: mesangial score ≤0.5 (M0) or >0.5 (M1), endocapillary hypercellularity absence (E0) or presence (E1), segmental glomerulosclerotic absence (S0) or presence (S1), tubular atrophy/interstitial fibrosis ≤25% (T0) or >25% (T1+T2), crescent absence (C0) or presence (C1+C2). Pa < 0.05 indicated comparison with IgAN-1; Pb < 0.05 indicated comparison with IgAN-2; Pc < 0.05 indicated comparison with IgAN-3.

Urinary protein extraction

The urine supernatants were precipitated by ethanol overnight at −20 °C. After centrifugation at 12,000 × g for 30 min, the pellets were resuspended in lysis buffer (7 M urea, 2 M thiourea, 0.1 M DTT, 50 mM Tris). The protein concentration of each sample was measured using the Bradford method. In the discovery phase, highly abundant proteins were removed from the urine protein samples using an Agilent Mars14 chromatography column following the manufacturer’s protocol (Agilent, Palo Alto, CA, USA). Equal amounts of total urine protein from each individual patient within one patient group were then pooled together. Finally, four pooled protein samples were formed from IgAN 1-3 and pMN patients for proteomic analysis.

Protein digestion and iTRAQ labelling

Each sample was digested using the filter-aided sample preparation method (FASP) (Wiśniewski et al., 2009). Protein samples were denatured with 20 mM DTT at 37 °C for 1 h and alkylated with 50 mM IAA in darkness for 45 min. Then, the samples were loaded onto filter devices with a cut-off of 10 kD (Pall, Port Washington, NY, USA) and centrifuged at 14,000g at 18 °C. After washing twice with UA (8 M urea in 0.1 M Tris-HCl, pH 8.5) and twice with 25 mM NH4HCO3, the samples were redissolved in 25 mM NH4HCO3 and digested with trypsin (enzyme to protein ratio of 1:50) at 37 °C overnight. The digested peptides were collected as a filtrate and desalted using Oasis HLB C18 cartridges (Waters, Milford, MA). The IgAN 1-3 and pMN samples were individually labelled with 114, 115, 116, and 117 4-plex iTRAQ chemistry according to the manufacturer’s protocol (ABsciex, Framingham, MA, USA).

2D-LC/MS/MS

2D-LC–MS/MS detection was performed using the methods described by Guo et al. (2015). The pooled mixture of iTRAQ-labelled samples was fractionated using a high-pH RPLC column from Waters (4.6 mm × 250 mm, Xbridge C18, 3 μm). The samples were loaded onto the column in buffer A1 (H2O, pH = 10) and then eluted under a gradient of 5–25% buffer B1 (90% ACN, pH =10) for 60 min with a flow rate of 1 mL/min. The eluted peptides were collected at fractions by minute and dried by vacuum evaporation for further redissolution. The 60 dried fractions were then resuspended in 0.1% formic acid and pooled into 20 samples, combining fractions by time intervals (combining 1, 21, 41 and 2, 22, 42 and so on). The resulting 20 fractions from the urinary peptide mixtures were analysed by LC–MS/MS.

Each fraction was analysed with a reversed-phase C18 capillary column (75 μm × 100 mm). An Orbitrap Fusion Lumos Tribrid mass spectrometer was used to analyse each mixed peptide fractions two times. A high-sensitivity mode was applied to acquire the MS data.

Western blot analysis

Two candidate proteins associated with glycosylation, NAGLU and GLA, were selected for western blot validation in individual urine samples from IgAN and pMN patients. Thirty micrograms of urinary protein from each sample were loaded onto 10% SDS‒PAGE gels and transferred to PVDF membranes. After blocking in 5% milk for 1 h, the membranes were incubated with primary antibodies at 4 °C overnight. Then, the membranes were washed with TBST four times and incubated with secondary antibodies (diluted 1:5,000 in 5% milk solution) for 1 h at room temperature. The blots of NAGLU and GLA were visualized using enhanced chemiluminescence reagents (Thermo Scientific, Waltham, MA, USA). The intensity of the blots was scanned with an ImageQuant 400TM Imager (GE Healthcare Life Sciences, Piscataway, NJ, USA) and the protein signals were quantified using the AlphaEaseFC system.

Parallel reaction monitoring (PRM) analysis

For PRM validation, the differentially expressed proteins from the LC‒MS/MS analysis were evaluated in a validation cohort of 64 IgAN patients. Each sample was analysed in schedule mode. The mixed samples were analysed for quality control (QC) to ensure the availability of data and the stability of the instrument signal for the PRM validation.

An Orbitrap Fusion Lumos Tribrid mass spectrometer, coupled with an EASY-nLC 1000 HPLC system, was used for the PRM analysis. An RP C18 self-packing capillary LC column (75 mm × 100 mm) was applied to separate the peptides. The elution gradient was 5–30% buffer B1 (0.1% formic acid, 99.9% ACN; flow rate, 0.5 ml/min), and peptides were eluted for 45 min. The MS data was acquired in PRM mode with the following parameters: full scans acquired (resolution: 60,000), MS/MS scans (resolution: 15,000), 32% collision energy, 30 s dynamic exclusion, 100–1,800 m/z MS/MS scan range, 100 ms scan time, isolation window at 4 Da, and a schedule window of 7 min.

The detailed parameters of PRM mode were as follows: data-dependent MS/MS scans per full scan with top-speed mode (3 s), full scans (resolution = 60,000) and MS/MS scans (resolution = 15,000), 38% HCD collision energy, charge-state screening (charge state of precursors: from +2 to +6), dynamic exclusion (exclusion duration 1:30 s), and maximum injection time (60 ms).

Data processing

Proteome Discoverer (Thermo Fisher, Waltham, MA, USA; version 2.1) was used for searching proteome data using the SwissProt human database (20,227 entries) and assuming trypsin digestion (http://www.ebi.ac.uk/swissprot/). The parent and fragment ion mass tolerances were 10 ppm and 0.05 Da, respectively. Carbamidomethyl cysteine was specified as a fixed modification, with two miscleavage sites accepted. Proteins were identified using a false discovery rate (FDR) lower than 1.0% and with at least two unique peptides identified. The acquired intensities were generally normalized across all runs. The normalization of the reference channels was also used to determine a 1:1-fold change. All normalization processing was achieved with medians in a multiplicative manner.

Skyline 21.1 was used for the selection of the suitable m/z precursor ion to m/z fragment ion transition for the PRM, in order to identify the candidate peptides of potential biomarkers. The required data was imported into the Skyline software, where the data for the PRM was chosen manually, and the resulting peptides from all samples were then exported. For each sample, the total ionic chromatography (TIC) of ions with +2 to +5 charge was obtained using the Progenesis software. Each peptide was then normalized by TIC strength to adjust for errors caused by the sample loading quality and MS signal intensity. The quantitation of the peptides was subsequently analysed.

Statistical analysis

Continuous data are presented as mean ± standard deviation or median. Categorical data are presented as absolute frequencies and percentages. The Kolmogorov–Smirnov test was performed to evaluate the normality of the distribution. For the quantitative data, one-way analysis of variance (ANOVA) was used to compare the differences for normally distributed data, and a non-parametric test was performed for non-normally distributed data. A chi-square test was used to compare the categorical variables among the groups. p values < 0.05 were considered statistically significant. The data was analysed with SPSS version 26.0 software (SPSS Inc., Chicago, IL, USA).

Receiver operating characteristic (ROC) curve analyses of all the significantly different proteins were performed using the Metaboanalyst software (http://www.metaboanalyst.ca). The combination ROC analysis used the linear SVM algorithm, and the area under the ROC curve (AUC) was calculated by separately comparing IgAN-2 and IgAN-3 with IgAN-1 during the validation phase to identify candidate markers.

Bioinformatics analysis

A clustering analysis was performed using the Mfuzz package in R to detect different clustering models of protein expression among the IgAN groups of different CKD stages in the discovery phase. We applied a pathway enrichment analysis to determine the different pathways of the proteins with statistical significance according to the KEGG (Kyoto Encyclopedia of Genes and Genomes) database. Protein interaction networks were constructed to demonstrate the potential associations among the different pathways. The data analysis was performed in the OmicsBean workbench (http://www.omicsbean.cn). An adjusted p value < 0.05 was considered statistically significant.

Results

Clinical and pathological characteristics of the patients

In this study, a total of 30 patients were included in the discovery phase, and another independent cohort was enrolled for validation, including 26 patients for western blotting, and 64 patients for PRM. The clinical data and pathological characteristics of the patients are summarized in Table 1. In the iTRAQ, WB, and PRM phases, there were significant differences in SCr and eGFR among the IgAN 1-3 and pMN patient groups. There were significant differences in the Oxford classification T-scores of the IgAN-3 group compared with the IgAN-1/2 groups in the discovery phase and significant differences in T-scores between the IgAN-3 and IgAN-1 groups in the validation phase (p < 0.05; Table 1).

Bioinformatics analysis of urine proteomes in IgAN and pMN patients

The workflow of the urine proteomics analysis is shown in Fig. 1. Urine samples of IgAN-1, IgAN-2, IgAN-3 and pMN patients were respectively pooled, and an antibody column was used to remove 14 high-abundance proteins from the urine samples prior to the proteomic analysis. A total of 747 urinary proteins were identified with at least two unique peptides in both technical replicates using 2D-LC‒MS/MS and iTRAQ quantification (Table S1). Western blotting of both alpha-N-acetylglucosaminidase (NGALU) and alpha-galactosidase A (GLA) was applied in IgAN and pMN patients. PRM was used for the targeted quantification of unique peptides from differentially expressed proteins (DEPs) in another IgAN 1-3 cohort.

Figure 1 The workflow of differential urinary proteome analysis, western blotting and parallel reaction monitoring validation.

A hierarchical clustering analysis of all urinary proteins found that there were different urine protein expression patterns in the IgAN 1-3 groups compared to pMN (Fig. 2A). The Mfuzz clustering analysis of 747 urinary proteins in the three IgAN subgroups is shown in Fig. 2B. Among the four clusters, Cluster 1, consisting of 120 urinary proteins, increased with IgAN progression, and the main enrichment pathways included complement and coagulation pathways and Staphylococcus aureus infection (Fig. 3A). The protein‒protein interaction (PPI) analysis showed that these three pathways are mainly related to complement C3, C5-9, complement lectin pathway (LP), mannan-binding lectin (MBL), and alternative pathway (AP) complement regulatory proteins (Fig. 3B).

Figure 2 Hierarchical clustering analysis of 747 urinary proteins from four pooled samples.

(A) Heatmap of urinary proteins in IgAN subgroups and pMN group (two technical replicates). The color bar from orange to blue represented the fold change from increasing to decreasing of the proteins in four groups. (B) Mfuzz correlation analysis of urinary proteins among IgAN subgroups. Proteins in cluster 1 were increased with IgAN progression; Proteins in cluster 2 were decreased with IgAN progression; Proteins cluster 3 (n = 142) and Cluster 4 (n = 291) showed no gradual changes with IgAN progression. Color bar represented Z-score change from −2 to 2.

Figure 3 Functional analysis of urinary proteins in cluster 1 and cluster 2 of IgAN patients.

(A) KEGG pathway enrichment analysis of proteins in cluster 1; (B) protein-protein interaction (PPI) networks were created for cluster 1 proteins; (C) KEGG pathway enrichment analysis of proteins in cluster 2; (D) PPI networks were created for cluster 2 proteins. In the KEGG enrichment analysis, the abscissa represented the percent of corresponding genes under each pathway classification (p value was noted in the bracket). The ordinate represented enrichment pathway. In the protein interaction networks, circle nodes represented for genes/proteins, rectangle for KEGG pathways. Blue solid lines represented inhibition; red solid lines represented activation; blue dotted lines represented KEGG pathways. The significance of the pathways represented by −log (p value) was shown by color bar with dark blue as the most significant. The color of circle nodes from red to blue represented the fold change from increasing to decreasing of the proteins.

The expression of 194 proteins in Cluster 2 gradually decreased with decreasing eGFR. A further pathway enrichment analysis showed that Cluster 2 was involved in cell adhesion molecules, ECM-receptor interactions, regulation of the actin cytoskeleton, and the PI3K-Akt signalling pathway (Fig. 3C). A PPI analysis demonstrated that collagen and fibronectin in these pathways were downregulated (Fig. 3D).

Both IgAN-1 and pMN patients in this study had similar eGFRs. We compared the urinary proteins of the IgAN-1 group with those of the pMN group according to a cut-off value of fold change >1.50 or <0.67, used to indicate differentially expressed proteins. There were 150 differentially expressed proteins between the IgAN-1 and pMN groups, 107 of which were upregulated. A clustering analysis of 107 proteins revealed that the main enrichment pathways were: regulation of actin cytoskeleton, focal adhesion, and protein digestion and absorption. The PPI analysis showed that structural proteins, such as collagen 4-6, nectin1, and nectin2, were more highly expressed in abundance in IgAN-1 than in pMN. Moreover, as shown in Table S1, 17 urinary proteins showed differential abundances between pMN and IgAN patients (15 proteins were expressed at higher levels in pMN than in IgAN1-3 patients).

Urinary complement pattern analysis in IgAN and pMN patients

Among the 747 proteins identified in urine samples, 27 were complement proteins. The urinary complement proteins were divided into five categories: the classical pathway (C1 s, C1rL, C1qR, C1 inhibitor SERPING1, C2 and C4), the lectin pathways (MBL, MASP1, MASP2 and Ficolin2), the AP complement regulatory proteins (CFB, CFD, CFH, CFHR2, CFHR4, CFI, DAF and properdin), the membrane attack complex (MAC) in the terminal pathways (C5, C6, C7, C8A, C8B, C8G, C9 and MCP (membrane cofactor protein)), and C3 (the full name of the proteins are shown in Table 2). The abundance of urinary C3, MAC, and complement regulatory proteins increased during the progression of IgAN. MAC compositions increased most significantly, while the classical pathway proteins did not increase with the progression of IgAN. For proteins in the LP of the complement cascade, as eGFR declined, MASP1 and MBL were upregulated, while MASP2 and Ficolin2 were downregulated. The abundance of five categories of urinary complement proteins was similar in IgAN-1 patients and pMN patients (Fig. 4, Table 2).

Table 2 The relative abundance of all urinary complements in IgAN 1-3 and pMN patients.

Complement	Uniprot	Gene name	Description	IgAN-1	IgAN-2	IgAN-3	pMN	Molecular weight (kDa)	
C3	P01024	C3	Complement 3	64	103	171	62	187.03	
LP	P11226	MBL2	Mannose binding protein	67	109	170	54	26.13	
Q15485	FCN2	Ficolin-2	161	76	69	94	33.98	
P48740	MASP1	Mannan-binding lectin serine protease 1	80	103	129	89	79.20	
O00187	MASP2	Mannan-binding lectin serine protease 2	148	59	56	137	75.65	
CP	P09871	C1S	Complement C1s	71	113	157	59	76.64	
Q9NZP8	C1RL	Complement C1r	144	62	68	126	53.46	
Q9NPY3	CD93	Complement component C1q receptor	97	75	108	120	68.52	
P05155	SERPING1	Plasma protease C1 inhibitor	155	69	80	96	55.12	
P06681	C2	Complement 2	52	103	202	44	83.21	
P0C0L4	C4A	Complement 4	59	76	221	44	192.65	
Complement regulatory proteins of AP	P00751	CFB	Complement factor B	63	92	182	63	85.48	
P00746	CFD	Complement factor D	102	81	107	110	27.02	
P08603	CFH	Complement factor H	75	85	180	60	139.01	
P36980	CFHR2	Complement factor H-related protein 2	65	81	198	56	30.63	
Q92496	CFHR4	Complement factor H-related protein 4	76	76	183	66	37.27	
P05156	CFI	Complement factor I	98	89	148	65	65.71	
P27918	CFP	Properdin	57	176	123	45	51.24	
P08174	CD55	Complement decay-accelerating factor	85	103	112	100	41.37	
MAC	P01031	C5	Complement 5	54	76	229	42	188.19	
P13671	C6	Complement 6	78	101	158	63	104.72	
P10643	C7	Complement 7	73	114	130	83	93.46	
P07357	C8A	Complement component C8 alpha chain	68	92	189	51	65.12	
P07358	C8B	Complement component C8 beta chain	59	85	198	59	67.00	
P07360	C8G	Complement component C8 gamma chain	72	91	175	62	22.26	
P02748	C9	Complement 9	64	67	206	63	63.13	
P15529	CD46	Membrane cofactor protein	79	105	127	89	43.72	
Note:

The complement proteins were divided into C3, CP (classical pathway), LP (lectin pathway), AP (alternative pathway) and MAC (membrane attack complex).

Figure 4 Comparison of urinary complement proteins in IgAN 1-3 and pMN patients.

CP, classical pathway; LP, lectin pathway; AP, alternative pathway; MAC, membrane attack complex.

Western blot validation of glycosylation-related proteins

Two proteins involved in glycosylation and glycan modification, alpha-N-acetylglucosaminidase (NAGLU) and alpha-galactosidase A (GLA), were further validated by western blot in another patient group, including IgAN 1-3 and pMN (Fig. 5, S2). The expression level of NGALU gradually decreased among IgAN 1-3 patients, and the expression level of NGALU in IgAN-3 patients was significantly lower than that in IgAN-1 and pMN patients (p < 0.05). The ratio of NAGLU abundances among the IgAN-1, IgAN-2, IgAN-3, and pMN groups in the MS experiment was 1:0.52:0.48:0.82, respectively. This result was consistent with the MS results. GLA also showed gradually decreased expression among IgAN 1-3 patients. This result was also consistent with its decreasing levels in the MS experiment (IgAN-1: IgAN-2: IgAN-3: pMN, 1:0.70:0.52:0.81). GLA also showed different expression levels between IgAN-3 and pMN patients (p < 0.05).

Figure 5 Western blotting validation of NAGLU and GLA in another patient cohort.

In the WB experiments, equal amount of urine protein from each individual sample was loaded. The validation set included IgAN-1 (n = 6), IgAN-2 (n = 8), IgAN-3 (n = 3) and pMN (n = 5) patients. Means and standard deviations were represented in the figure, and a nonparametric test were used to analysis the data. * Indicates a p value < 0.05.

PRM validation of IgAN markers

To validate the iTRAQ results, 64 urinary samples were collected, including 14 from IgAN-1 (eGFR: >90 ml/min/1.73 m2), 29 from IgAN-2 (eGFR: 60–90 ml/min/1.73 m2), and 21 from IgAN-3 (eGFR: <60 ml/min/1.73 m2).

To evaluate LC‒MS/MS system stability, a mixture of all urine samples was pooled as quality control (QC) samples. The correlation map of QC samples showed that QC samples were highly positively correlated with an average Pearson correlation coefficient of 0.966 (Fig. S3). These results showed the repeatability and stability of the LC‒MS/MS platform.

The PRM analysis validated 17 peptides that corresponded to 10 proteins which showed the same trend as the iTRAQ results, including two upregulated proteins and eight downregulated proteins (Fig. 6 and Table 3). CFB and C8A were verified to be upregulated. Among the downregulated proteins, SERPINA5 (plasma serine protease inhibitor), CD58 (lymphocyte function-associated antigen 3), and mucosal addressin cell molecule 1 (MAdCAM1) presented a gradually downward trend with the progression of IgAN, indicating these proteins might be potential markers for monitoring the progression of IgAN. DANSE2 (Deoxyribonuclease 2) and AXL (Tyrosine-protein kinase receptor UFO) significantly decreased only in IgAN-3 (Fig. 6), indicating these proteins might be associated with the progression of IgAN.

Figure 6 The PRM results of validated differential proteins.

IgAN-1: eGFR >90 ml/min/1.73 m2, IgAN-2: eGFR: 60–90 ml/min/1.73 m2, IgAN-3: eGFR: 30–60 ml/min/1.73 m2. * Indicates a p value < 0.05, ** indicates a p value < 0.01, *** indicates a p value < 0.001, **** indicates a p value < 0.0001, ns indicates a p value > 0.05. CFB, complement factor B; C8A, complement component C8 alpha chain; SERPINA5, plasma serine protease inhibitor; CD58, lymphocyte function-associated antigen 3; MADCAM1, mucosal addressin cell adhesion molecule 1; AXL, tyrosine-protein kinase receptor UFO; ZG16B, zymogen granule protein 16 homolog B; EGF, pro-epidermal growth factor; DNASE2, deoxyribonuclease-2-alpha; LRG1, leucine-rich alpha-2-glycoprotein.

Table 3 Validated proteins by parallel reaction monitoring.

Accession	Gene name	Peptides	iTRAQ
fold change	PRM
p value	PRM
fold change
(IgAN-2/1)	AUC
(IgAN-2/1)	PRM
fold change (IgAN-3/1)	AUC
(IgAN-3/1)	
P07357	C8A	YNPVVIDFEMQPIHEVLR	2.344	0.004	1.579	0.788	1.523	0.776	
P00751	CFB	LLQEGQALEYVCPSGFYPYPVQTR	2.181	0	1.851	0.899	1.345	0.857	
P19256	CD58	VAELENSEFR	0.920	0.003	0.901	0.764	0.881	0.844	
P05154	SERPINA5	GFQQLLQELNQPR	0.850	0.002	0.932	0.714	0.915	0.854	
EDQYHYLLDR	
AAAATGTIFTFR	
O00115	DNASE2	YLDESSGGWR	0.663	0.004	0.978	0.544	0.837	0.765	
ALINSPEGAVGR	
Q96DA0	ZG16B	YFSTTEDYDHEITGLR	0.645	0.01	0.895	0.764	0.896	0.793	
Q13477	MADCAM1	GLDTSLGAVQSDTGR	0.631	0	0.829	0.783	0.788	0.857	
LPGLELSHR	
P01133	EGF	IYFAHTALK	0.660	0.003	0.927	0.761	0.926	0.816	
LIEEGVDVPEGLAVDWIGR	
P02750	LRG1	TLDLGENQLETLPPDLLR	0.496	0.009	0.943	0.791	0.969	0.684	
DGFDISGNPWICDQNLSDLYR	
GQTLLAVAK	
P30530	AXL	TATITVLPQQPR	0.462	0.008	0.948	0.628	0.906	0.786	
Note:

C8A, complement component C8 alpha chain; CFB, complement factor B; CD58, lymphocyte function-associated antigen 3; SERPINA5, plasma serine protease inhibitor; DNASE2, deoxyribonuclease-2-alpha; ZG16B, zymogen granule protein 16 homolog B; MADCAM1, mucosal addressin cell adhesion molecule 1; EGF, pro-epidermal growth factor; LRG1, leucine-rich alpha-2-glycoprotein; AXL, tyrosine-protein kinase receptor UFO.

The diagnostic values of 10 proteins were evaluated using ROC analyses. Seven proteins had an AUC >0.75 when comparing IgAN-2 with IgAN-1 and nine proteins had an AUC >0.75 when comparing IgAN-3 with IgAN-1 (Table 3). Meanwhile, the combination of CFB and MAdCAM1 showed an AUC of 0.963 (95% CI [0.856–0.997])/0.915 (95% CI [0.770–0.982]) in distinguishing IgAN-2/3 from IgAN-1 (Figs. 7A and 7B).

Figure 7 The analysis of PRM results.

(A) Receiver operating characteristic (ROC) analysis of IgAN-2 and IgAN-1; (B) ROC curve analysis of IgAN-3 and IgAN-1; (C) heatmap of correlations between PRM results and clinical tests. Color bar represented the correlations from −0.6 to 0.6. eGFR, estimated glomerular filtration rate; C3, complement component 3; C4, complement component 4; 24hUP, 24-h urinary protein; BUN, blood urine nitrogen; MAP, mean arterial pressure; M, mesangial hypercellularity; E, endocapillary cellularity; S, segmental sclerosis; T, interstitial fibrosis/tubular atrophy; C, crescents.

We further analysed whether these potential biomarkers correlated with the clinical characteristics of IgAN. A Spearman’s correlation analysis showed that the urinary excretion of CFB and C8A was negatively correlated with eGFR decline, while the other eight proteins had a positive correlation with eGFR. The abundance levels of C8A, SERPINA5, and AXL also had a strong correlation with 24-h urinary protein (24hUP; |r| > 0.3). Seven of the 10 proteins, including CFB, C8A, DNASE2, SERPINA5, AXL, and EGF (Pro-epidermal growth factor), were also significantly associated with MEST-C scores (|r| > 0.2) and were most significantly correlated with the Oxford classification T-score, which indicates interstitial fibrosis/tubular atrophy in IgAN (Fig. 7C).

Discussion

The aim of this study was to evaluate the urine proteomics of patients with different stages of IgAN, especially the complements and complement-related proteins.

Activation of the complement pathway plays an important role in the pathogenesis of IgAN. The development of targeted complement inhibitors in glomerular diseases (Andrighetto et al., 2019) requires a better understanding of how the complement pathway is involved in IgAN progression. In this study, we found that the urinary complement proteins C3, MAC, and the complement regulatory proteins increased with the progression of IgAN, indicating AP activation is involved in the progression of IgAN.

In a previous urinary proteomic study of healthy individuals, Shao et al. (2019) identified 1,872 quantifiable proteins using 2D LC‒MS/MS with the label-free quantitative method, including 26 of the 27 complements reported in our study. In addition, Zhao et al. (2017) identified all 27 complements found in our study from pooled urinary samples of healthy individuals using the same proteomic-based technology. In IgAN, IgA is mostly codeposited with C3 in the mesangium. MBL (Roos et al., 2006), properdin, and CFH deposition (Miyazaki et al., 1984) were also found in the mesangium. A recent proteomic study of microdissected glomeruli from progressive IgAN cases revealed significantly increased MAC proteins, FHR5, and FHR2, compared with patients with a stable clinical status (Paunas et al., 2017). In our study, 20 urinary complement proteins were found to be consistent with those previously found in the microdissected glomeruli; notably, we found eight differentially expressed proteins of LP and complement regulatory proteins that have not been identified in the microdissected glomeruli of IgAN patients (Table S2).

Previous studies suggest that the overactivation of AP-dependent complement amplification could promote the formation of the MAC and lead to kidney injury in IgAN. According to the iTRAQ results in this study, CFH, CFHR2 and CFHR4 were found to have the highest expression in IgAN-3 (CFH: (IgAN1: IgAN2: IgAN3) = 1: 1.14: 2.41; CFHR2: (IgAN1: IgAN2: IgAN3) = 1: 1.25: 3.06, CFHR4: (IgAN1: IgAN2: IgAN3) = 1: 1: 2.41). CFH is the main negative regulator of AP by binding with C3b and inhibiting the formation of C3 convertase, which normally functions to prevent the excessive activation of the complement system. The CFHRs are sequentially similar to CFH and could compete with CFH for C3b binding. These proteins have been studied as potential risk factors for IgAN (Medjeral-Thomas, Lomax-Browne & Pickering, 2018). Previous studies have reported that high levels of CFHR1, CFHR3, and CFHR5 could increase C3b binding capacity and AP activation, resulting in complement-mediated injury in IgAN (Lukawska, Polcyn-Adamczak & Niemir, 2018; Cserhalmi et al., 2019). Although there are fewer reports on urinary CFHR2 or CFHR4 in IgAN, the functions of these two proteins indicate that they may also facilitate complement system activation. In vitro studies have demonstrated that CFHR4 initiates AP activation through the formation of active C3 convertases on CFHR4-bound C3b (Hebecker & Jozsi, 2012). Compared with other CFHRs, CFHR2 has a lower affinity to compete with CFH in binding C3b or C3d but can form homologous dimers or heterodimers with CFHR1 and CFHR5 (Lukawska, Polcyn-Adamczak & Niemir, 2018). Complexes of CFHR1, CFHR2, or CFHR5 have been reported to increase the avidity for the C3b, iC3b, and C3dg ligands and enhance competition with complement Factor H (CFH), eventually promoting AP activation on certain surfaces (Goicoechea de Jorge et al., 2013; Tortajada et al., 2013). The significant increase in CFH, CFHR2, and CFHR4 in IgAN-3 may reflect these potential effects in the progression of IgAN.

There have been few studies on changes in serum or urinary LP proteins of the complement cascade in IgAN patients. In this study, we identified four urinary LP proteins in IgAN patients. Unlike MBL and MASP1, urinary MASP-2 and ficolin-2 gradually decreased with IgAN progression. MASP-2 could directly cleave C3 while bypassing the usual sequence to activate C4 and/or C2, therefore playing an important role in the LP activation of the complement system (Yaseen et al., 2017). A pilot study recently showed that nasoplimab, a MASP-2 inhibitor, could reduce proteinuria and help maintain renal function in IgAN patients, suggesting that MASP-2 activation may be involved in the pathogenesis of IgAN (Lafayette et al., 2020). The reasons for the downregulation of MASP-2 and ficolin-2 in progressive IgAN patients are not fully understood. A British study compared levels of plasma LP proteins between stable and progressive IgAN patients and found that M-ficolin, L-ficolin, MASP1, and MAp19 were increased and MASP3 was decreased in progressive IgAN patients, and there was no difference in MASP-2 between IgAN patients and healthy controls (Medjeral-Thomas et al., 2018). These changes are thought to be secondary to LP activation, with MASP2 consumption at sites of disease activity. Further studies are required to assess these changes in urinary MASP2 and ficolin-2 in progressive IgAN patients. In this study, the urinary complement profile of pMN patients with eGFR >90 ml/min/1.73 m2 was analysed as a control. Primary membranous nephropathy is caused by autoantibodies that bind to antigens expressed by podocytes. These autoantibodies, which are predominantly of the IgG4 subclass, mainly activate the LP and AP (Ma, Sandor & Beck, 2013). Therefore, the abundance of urinary complement proteins in pMN patients with normal renal function was similar to that observed in IgAN-1 patients (Fig. 4). A total of 17 differential urinary proteins between pMN and IgAN patients were also identified in the discovery phase, and their significance should be further investigated.

The proteins in Cluster 1 increased with eGFR reduction, indicating that these proteins are closely associated with disease severity in IgAN and could be used as non-invasive biomarkers of disease progression. In Cluster 1, the KEGG pathway analysis showed that the complement and coagulation cascade pathways, immune disease pathway, and Staphylococcus aureus infection were significantly enriched. A cohort study including 116 IgAN patients and 122 non-IgAN patients revealed that 68% of IgAN patients presented with Staphylococcus aureus antigen, while non-IgAN patients had no Staphylococcus aureus antigen deposition (Koyama et al., 2004). Our study revealed that the Staphylococcus aureus pathway was significantly activated with IgAN progression and that Staphylococcus aureus interacted with MBL, C3, C4, and C5. A recent study (Kurokawa, Takahashi & Lee, 2016) reported that Staphylococcus aureus could activate LP by acting on MBL through the teichoic acid on the surface of the glycol polymer wall.

Cluster 2 proteins decreased with eGFR progression. Further pathway analysis revealed that the first three enrichment pathways were: cell adhesion molecules, focal adhesion, and the PI3K-Akt signalling pathway. The mesangial deposition of IgA and ECM expansion have been thought to be the initiating events in IgAN pathogenesis. Our results showed that the PI3K-Akt signalling pathway was significantly enriched at the early stage of IgAN. A gene array study (Cox et al., 2010) proposed that differentially expressed genes between IgAN patients and healthy volunteers were mainly involved in the PI3K-Akt signalling pathway, which is strongly related to cell proliferation.

Two glycosylation-related enzymes, NAGLU and GLA, were also assessed by western blotting in IgAN 1-3 and pMN patients, and the WB results were consistent with the iTRAQ quantitative proteomics results. NAGLU is a type of glycosidase that is involved in the degradation of the heparan sulfate (HS) polysaccharide by removing terminal N-acetylglucosamine (GlcNac) residue (Uhlen et al., 2015). It has been reported that HS can promote fibrosis in the kidney (Furini & Verderio, 2019). HS deficiency in mice was demonstrated to prevent the profibrotic crosslinking of the extracellular matrix and recruitment of TGF-β1 (Talsma et al., 2018; Scarpellini et al., 2014), which was positively correlated with fibrotic area in the renal tissue of IgAN patients (Yang et al., 2021). Therefore, a decline in NAGLU could result in the accumulation of heparan sulfate (HS) (Mohammed & Fateen, 2019) and may contribute to IgAN progression.

Alpha-galactosidase (GLA) is a lysosomal glycosidase and can catalyse the removal of α-linked galactose from oligosaccharides, glycoproteins, and glycolipids during the salvage pathway of galactose (Lombard et al., 2014; Coelho, Berry & Rubio-Gozalbo, 2015). In humans, the deficiency of functional GLA could result in the accumulation of galactosylated substrates (Gb3) in tissues, leading to Fabry disease, which is often accompanied by a series of visceral diseases, including renal disease (Kok et al., 2021). According to previous reports, IgAN and Fabry disease may coexist (Whybra et al., 2006; Sun et al., 2022; Lerner et al., 2022). However, the mechanism of GLA in IgAN is still unknown. The WB results of NAGLU and GLA were consistent with the iTRAQ quantitative proteomics results. This is the first study showing the changes in these two proteins in the urine of IgAN patients, although the clinical significance of these proteins needs to be further explored.

Among the DEPs validated in PRM, the increase in CFB and C8A, consistent with the iTRAQ results, confirmed the activation of the complement system in IgAN (Fig. 6). CFB is an essential component of the alternative pathway, providing an active subunit associated with C3b to form the C3 convertase (Alfakeeh et al., 2017). Previous studies have revealed increasing CFB or complement factor Bb (a fragment of CFB) in the circulation of patients with pMN or IgAN (Zhang et al., 2019). In our study, CFB exhibited a negative correlation with eGFR but a positive correlation with 24-h urine protein, which supports the involvement of CFB and AP activation in IgAN progression. C8A, as a component of the MAC, was significantly increased in IgAN-2 and IgAN-3 compared with IgAN-1 in our study (Fig. 6). The formation of MAC or C5b-9 was reported to induce glomerular injury, inflammation, and fibrosis (Koopman et al., 2020). C5b-9 serum was also evaluated as a prognostic marker for IgAN. Several studies have demonstrated more intense and diffuse C5b-9 deposition in the glomerulus and tubules of IgAN patients than in healthy controls (Koopman et al., 2020; Maixnerova et al., 2016; Yu et al., 2022). Urine-based proteomics studies have also suggested an increase in urinary C5b-9 levels in patients with IgAN. Urinary C5b-9 levels were found to be associated with disease severity and inversely correlated with eGFR (Yu et al., 2022; Abe et al., 2001; Neuen et al., 2021). The observed increase in CFB and C8A confirmed that the AP and MAC might participate in pathological processes during the progression of IgAN.

The combination of CFB and MAdCAM-1 was able to distinguish different levels of IgAN in this study. MAdCAM-1 is a transmembrane protein with a complex structure that can combine three Ig domains and a mucin-like region between Ig domains 2 and 3. There is also an IgA-like Ig domain in the MAdCAM-1 molecule, which could be correlated with its expression in the area where most IgA antibodies are secreted (Shyjan et al., 1996). MAdCAM-1 is involved in the migration and recruitment of leukocytes to the site of inflammation in chronically inflamed tissues (Ando et al., 2007). Previous studies have suggested that the tissue expression of adhesion molecules in IgAN reflects continuous inflammatory renal activity, including vascular cell adhesion molecule 1 (VCAM-1) and intercellular adhesion molecule 1 (ICAM-1) (Mrowka, Heintz & Sieberth, 1999; Wen et al., 2022). However, few studies have demonstrated the association between MAdCAM-1 and IgAN. A previous study detected soluble MAdCAM-1 in the serum and urine of healthy donors, and the expression levels of MAdCAM-1 were similar to those of CAM, ICAM-1, and VCAM-1 (Leung et al., 2004). The decline of MAdCAM-1 in the urine of IgAN patients should be further studied as a potential marker for IgAN progression.

After removing 14 highly abundant proteins in urine samples prior to quantitative proteomic analysis, this study identified abundant complements and complement-related proteins in the urine of IgAN patients in different CKD stages. To our knowledge, this is the first study to identify an abundance of complements in the urine to IgAN patients. This is also the first study to demonstrate alterations in the urinary complement proteins of AP and LP. Among these complements, the major component involved in IgAN progression was the MAC.

This study has several limitations. First, this was a single-centre study with a limited number of patients. Thus, a large-scale analysis, including healthy controls, IgAN patients, and patients with other related chronic kidney diseases, is needed to confirm the conclusions of this study. Second, although CFB and C8A were validated by PRM, more related complements, such as CFHR2 and CFHR4, need to be validated. Immunohistochemistry studies in the renal tissues of IgAN patients would also help to verify the results of this study.

In conclusion, complement components were found in IgAN urine and increased with the progression of IgAN. These findings indicate that the urine proteome may reflect changes in IgAN progression. In addition, a combination of CFB and MAdCAM-1 may be used as a potential urinary biomarker for monitoring the development of IgAN. The findings of our study provide useful information for future IgAN research and improve the understanding of the pathogenesis of IgAN.

Supplemental Information

Supplemental Information 1 Identification and quantitation details of urinary proteins in IgAN and pMN patients.

Click here for additional data file.

Supplemental Information 2 Differential complements between glomeruli and urine of IgAN patients.

Click here for additional data file.

Supplemental Information 3 Functional analysis of urinary proteins upregulated in IgAN-1 patients compared to pMN patients.

(A) KEGG pathway enrichment analysis of 107 upregulated proteins in IgAN-1 compared to pMN. (B)The PPI networks were created for 107 upregulated proteins in IgAN-1 compared to pMN.

Click here for additional data file.

Supplemental Information 4 The raw image of western blotting of NAGLU and GLA in another patient cohort.

Click here for additional data file.

Supplemental Information 5 Correlation of QC samples in PRM.

Click here for additional data file.

Supplemental Information 6 The raw data of iTRAQ and PRM.

Click here for additional data file.

Supplemental Information 7 Raw data for Figure 5-uncropped western blots.

Click here for additional data file.

Supplemental Information 8 Raw data for Table 1.

Click here for additional data file.

Supplemental Information 9 Numeric data for WB result.

Click here for additional data file.

We thank Ying Sun, Siqian Li, and Liling Lin for collecting specimens and Zhengguang Guo and Wei Sun for their great help in analysing the experimental data.

Additional Information and Declarations

Competing Interests

Author Contributions

Data Availability

The authors declare that they have no competing interests.

Xia Niu analyzed the data, prepared figures and/or tables, authored or reviewed drafts of the article, and approved the final draft.

Shuyu Zhang performed the experiments, analyzed the data, prepared figures and/or tables, authored or reviewed drafts of the article, and approved the final draft.

Chen Shao conceived and designed the experiments, authored or reviewed drafts of the article, and approved the final draft.

Zhengguang Guo conceived and designed the experiments, prepared figures and/or tables, and approved the final draft.

Jianqiang Wu performed the experiments, prepared figures and/or tables, and approved the final draft.

Jianling Tao conceived and designed the experiments, authored or reviewed drafts of the article, and approved the final draft.

Ke Zheng performed the experiments, authored or reviewed drafts of the article, and approved the final draft.

Wenling Ye analyzed the data, prepared figures and/or tables, and approved the final draft.

Guangyan Cai conceived and designed the experiments, authored or reviewed drafts of the article, and approved the final draft.

Wei Sun conceived and designed the experiments, authored or reviewed drafts of the article, and approved the final draft.

Mingxi Li conceived and designed the experiments, authored or reviewed drafts of the article, and approved the final draft.

The following information was supplied regarding data availability:

The raw data for LC-MS/MS analysis is available at iProX: IPX0006078000, PXD040799.

https://www.iprox.cn/page/project.html?id=IPX0006078000.

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
