# Peer review of "Urinary complement proteins in IgA nephropathy progression from a relative quantitative proteomic analysis"

_PeerJ, doi:10.7717/peerj.15125_

## Round 0.1 · original submission · Major Revisions

The reviewers identified some major flaws in the study design so fundamental changes are required. Please carefully resolve these (especially addition of a healthy control group, exclusion of lupus nephritis subjects, and the independent validation of the urinary complement's molecules) in order for your manuscript to be taken into consideration.

·

Basic reporting

The manuscript entitled “Urinary complement proteins in IgA Nephropathy progression from a relative quantitative proteomic analysis” by Shuyu Zhang et al. investigated the complements during IgA nephropathy (IgAN) progression by quantitative proteomic approach. They have shown that C3, the membrane attack complex (MAC) and the complement regulatory proteins of the alternative pathway (AP), MBL and MASP1 of the lectin pathway (LP) were increased along IgAN progression. They validated the expression of α-N-acetylglucosaminidase (NAGLU) and α-galactosidase A (GLA) in another cohort using immunoblotting.

Experimental design

1. In Figure 4, how did the authors determine statistical significance between the compared groups? It was not labeled in the figure.
2. In the raw blots of Figure 5, loading controls such as β-actin and GAPDH were missing. How did authors quantify the blots?

Validity of the findings

To support the role of urinary complements as biomarkers for evaluating IgAN progression, these proteins should be validated in an independent cohort. It is not sufficient to only detect NAGLU and GLA by Western blot.

Additional comments

1. “IgA nephropathy (IgAN) is one of the leading causes of ESRD”. What does ESRD stand for? The full name should be spelled at the first time it was mentioned.
2. Quality of Figure 2, 3 could be improved. Do authors have images with higher resolution?

Reviewer 2 ·

Basic reporting

The present manuscript is unambigous, and English language requires minor revision.
References are adequate, and the paper is well structured.
The topic is relevant.

Experimental design

I believe that the topic is original.
However, I that the design should be improved.
We lack of a normal control group. This is key. Authors must add at least 10 healthy patients.
Moreover, in the pathologic disease group, I would delete the lupus nephritis patients, as in IgAN the classical pathway is potentially not affected, and is a secondary cause of glomerulnephritis. I can not find any reason to include these patients. In primary nephropathy, the classiccal and the alternate pathway are involved.

What is the "dicovery" phase?. It must be defined in the Methods section.

Please refer to "complement components or molecules" and not to "complements".

Authors explanation about GLA findings may not be so accurate. Was the enzime galactosyltransferase approached?. If not, why was the reason?.

Please explain the causes by which CFHR2 and 4 and not 1,3 and 5 (which protect against IgAN) may stimulate IgAN progression.

The "n" of the study is not depicted in the abstract. Abbreviations as iTRAQ and ESRD must be explained in the Abstract or any other one at the first time it appears in the paper (WB, etc).

Validity of the findings

The impact of complement biomarkers on the pathophysiology of IgAN is high.
The statistics are correct form my point of view.
Conclusions could be better addressed.

Additional comments

I think that it is key to add a healthy control group.

Reviewer 3 ·

Basic reporting

The authors should mention to the readers why NAGLU and GLA are used as the validation protein in the introduction part and provide references.

The authors should justify their reasoning to categorize the patients into 4 groups in the discovery phase but 5 groups in the validation phase.

In the supplementary Western blot figure, the authors should include a internal control protein and also be consistent with the group name in this figure and figure 5 to make it easier to read, for example, A1-1 may be better named lgAN-1_1.

Experimental design

The major issue for this manuscript is the small sample size in each group and the pooled sample method that totally concealed the big variations, which is quite usual in clinical settings, in the protein level among patients. Since a statistical analysis is not possible here, the authors should be cautious with their conclusion draw from the analysis in figure 4.

Validity of the findings

The authors need to give better explanation to their conclusion "The results of WB validation were consistent with the results based on quantitative proteomics. The validation experiments showed the reliability of the iTRAQ quantitative results in the discovery period" since they have not provide any evidence to suggest a link between NAGLU/GLA and the complement pattern analysis.

---

## Round 0.2 · Minor Revisions

Please explain if anti-PLA2R blood level was measured. If not, add the attribute "presumable" to the definition of primary MN in the Methods section.

·

Basic reporting

Authors have successfully responded to my comments.

Experimental design

None.

Validity of the findings

None.

Additional comments

None.

Reviewer 2 ·

Basic reporting

The English language needs major polishing.
References are all right.
The design has been improved snce last submission
However I am concerned about the definition of primary membranous nephropathy:
Authors state: "The pMN was diagnosed by renal pathology (light microscopy, immunofluorescence and electron microscopy). Secondary MN due to infection, malignant neoplasms, rheumatologic disease, drugs, and other systemic causes was ruled out."
This is incorrect. pMN is defined by the presence of a circulatimg antibody, most commonly antiPLA2R or less frequently, thromobospondin-7, semaphorin, etc.
So, was antiPLA2R measured?. If not, pMN cannot be defined as such.

Experimental design

It is an original primary reseac¿rch, questions addressed and improved compared to previously submitted version. It follows techinical and ethical standards.
The methodology is appropriate.

Validity of the findings

My main concern is still the control group. Mainly, the way pMN was diagnosed. Were antibodies performed?. The way pMN is defined is incorrect. Please address
.Alternatively, a healthy control group could be added.

Additional comments

Nothing else to be added. All other previous insights have been addressed.

Reviewer 3 ·

Basic reporting

The authors have made great effort to significantly improve the wording, grammar, quality of data and interpretation of results that have basically addressed all of this review's concerns on the manuscript and it is this reviewer's opinion that it is now ready to be published.

Experimental design

N/A

Validity of the findings

N/A

Additional comments

N/A

---

## Round 0.3 · Minor Revisions

Dear authors,

Thank you for your documented lesson of Nephrology. However, I am sure that all the data you presented in the reply letter are well known by the reviewers and editor.

The request made in the second round of revisions was simple: please add the information about the lack (or not) of anti-PLA2R measurement and explain that the diagnosis of primary MN was made primary by clinical exclusion of main secondary causes. The biopsy only diagnoses the histological pattern (membranous), but the distinction primary/secondary is not possible with certainty (only histological hints might be obtained).

Therefore, in order to be fully transparent, you should accept the reviewer and editor's request to allay the shadow of doubt on the certainty of primary MN diagnosis by adding the statement about the lack of anti-PLA2R in addition to the already added paragraph.

---

## Round 0.4 · accepted · Accept

The authors have resolved all the previous concerns raised by the reviewers and editor.